# Plant Growth-Promoting Rhizobacteria (PGPR): A Rampart against the Adverse Effects of Drought Stress

Naoual Bouremani [1], Hafsa Cherif-Silini [1], Allaoua Silini [1], Ali Chenari Bouket [2], Lenka Luptakova [3], Faizah N. Alenezi [4], Oleg Baranov [5] and Lassaad Belbahri [6,*]

1. Laboratory of Applied Microbiology, Department of Microbiology, Faculty of Natural and Life Sciences, University Ferhat Abbas of Setif, Setif 19000, Algeria
2. East Azarbaijan Agricultural and Natural Resources Research and Education Centre, Plant Protection Research Department, Agricultural Research, Education and Extension Organization (AREEO), Tabriz 5355179854, Iran
3. Department of Biology and Genetics, University of Veterinary Medicine and Pharmacy in Košice, 04181 Kosice, Slovakia
4. Marine Biodiscovery Centre, Department of Chemistry, University of Aberdeen, Old Aberdeen, Aberdeen AB24 3UE, UK
5. Department of Biological Sciences, National Academy of Sciences of Belarus, 220072 Minsk, Belarus
6. Laboratory of Soil Biology, University of Neuchatel, 11 Rue Emile Argand, CH-2000 Neuchatel, Switzerland
* Correspondence: lassaad.belbahri@unige.ch

**Abstract:** Abiotic stress significantly limits plant growth and production. Drought, in particular, is a severe constraint that affects growth and limits agricultural productivity on a global scale. Water stress induces in plants a set of morpho-anatomical (modification of root and leaf structure), physiological, and biochemical (relative water content, membrane stability, photosynthesis, hormonal balance, antioxidant systems, and osmolyte accumulation) changes mainly employed to cope with the drought stress. These strategies allow the plant to overcome the unfavorable period of limited water availability. Currently, a promising alternative is available to improve plant growth and tolerance under drought conditions. The use of osmotolerant plant growth-promoting rhizobacteria (PGPR) as inoculants can alleviate water stress by increasing the water use efficiency of the plant. The PGPR improve the tolerance of plants to drought, through changes in the morphology and architecture of the root system, production of phytohormones, extracellular polysaccharides, ACC 1-(aminocyclopropane-1-carboxylate) deaminase, volatile chemicals, and osmolyte accumulation. They may also enhance the antioxidant defense system and induce transcriptional regulation of stress response genes. This review addresses the effects of stress on plant growth, adaptation, and response to drought conditions and discusses the significant potential of PGPR to modulate the physiological response against water scarcity, ensuring plant survival and improving the resistance and growth of agricultural crops.

**Keywords:** water stress; PGPR; osmotolerance; phytohormones; antioxidants

## 1. Introduction

Currently, climate change, exponential population growth and increased demand for nutrients are among the major challenges facing agriculture on a global scale. According to the most recent United Nations (UN) estimate, the world's population is currently 7.3 billion, and is expected to reach 9.7 billion by 2050 [1,2]. Rapid population growth is accompanied by the loss of fertile agricultural land to human occupation, land degradation, and increased anthropogenic activity, such as industrialization, urbanization, and deforestation. Changes in land use patterns result in the release of greenhouse gases. Continued increases in air temperature and atmospheric $CO_2$ levels ultimately alter the precipitation pattern and distribution, accelerating global climate change [2,3]. Global warming is unfortunately a reality already measured and confirmed in recent years in many countries.

In 2015, the European Environment Agency warned about the disruption of agricultural cycles in continental Europe. Rising temperatures with little or no precipitation increase evapotranspiration. A decrease in groundwater levels or water retention by soil particles causes soil drying and salinization [3,4].

Drought affects crop growth and development, as well as agricultural yields. The increasing frequency and intensity of drought from year to year are responsible for a dramatic drop in food production. Drought stress has been projected to cause a 10% decline in grain production over the past half century, with projections of productivity losses in more than 50% of arable land by 2050 [5]. The International Food Policy Research Institute (IFPRI) has found a sharp decline in most of the world's major agricultural crops such as wheat, corn, and rice. According to reports published between 1980 and 2015, water scarcity has contributed to a 21% reduction in wheat yields and a 40% reduction in corn production worldwide [6,7].

Water is the most important determinant of crop growth and production, and thus is an important influence on species distribution and evolution [8,9]. Water accounts for approximately 80–90% of total plant biomass. It is essential for almost all physiological processes and is a primary source of nutrition and metabolite supply [9,10]. Water deficit occurs when the plant's water requirements cannot be fully met, which occurs when the amount of water transpired exceeds the amount of water absorbed by the roots. This affects a variety of morphological, physiological, biochemical, ecological, and molecular characteristics and processes in plants. Lack of water is a major source of stress for plants; it reduces the total development of plants by decreasing photosynthetic activity, hormone synthesis, membrane integrity, etc. [11].

The need to protect, improve tolerance and increase productivity of staple crops such as wheat (*Triticum turgidum* L.), rice (*Oryza sativa* L.), maize (*Zea mays* L.), and vegetables under water deficit conditions compels researchers to move towards the development of several strategies such as the development of drought resistant cultivars, molecular breeding, genetic engineering, and the use of nanoparticles, film culture, superabsorbent hydrogels, and biochar. However, most of these procedures are time consuming, costly, laborious, and unpopular in some places with modest benefits [12,13].

A promising alternative to improve plant growth and tolerance under drought circumstances is the use of PGPR (plant growth-promoting rhizobacteria) as bacterial inoculants. In recent years, numerous studies have shown that the use of beneficial microorganisms can increase a plant's resistance to abiotic stresses such as drought, salinity, nutrient deficiency, and metal contamination. These beneficial bacteria adapted to osmotic stress improve plant development and stress tolerance through various direct and indirect means. Therefore, the use of a bacterial inoculant offers an alternative and economically viable strategy to support food security under drought conditions [11,14].

This review highlights the current knowledge of the effect of drought on plant growth, the mechanisms of plant tolerance to water deficit, and the use of PGPR to improve crop tolerance, growth, and productivity under water shortage conditions.

## 2. Effects of Water Stress on Plant Growth

Water stress is one of the most prevalent abiotic factors affecting crop growth and productivity. Drought is considered a multidimensional stress [13]. It influences the normal growth and development of plants by affecting their water potential and cell turgor. Alterations and changes in morphological, physiological, biochemical, and even molecular properties of plants are observed as a result of disruption of water retention and reduction in its use efficiency [15] (Table 1).

**Table 1.** Impact of water stress on plant growth and development.

| Plant Species | Effectiveness of Stress | Reference |
|---|---|---|
| Wheat (*Triticum aestivum* L.) | -Inhibition of seed germination.<br>-Reduction in seed germination rate, coleoptile weight length, and seed radicle length. | [16] |
| Rice (*Oryza sativa* L.) | -Limitations in photosynthesis, stomatal conductance, transpiration rate, maximum electron transport rate, and carboxylation efficiency. | [17] |
| Potato (*Solanum tuberosum* L.) | -Decrease in relative water content (RWC), photosynthetic rate, and chlorophyll a, b content.<br>-Increase in the content of electrical conductivity and the content of MDA, absissic acid (ABA), proline, soluble sugars, and polyamines. | [18] |
| Alfalfa (*Medicago sativa*) | -Decrease in photosynthesis rate, chlorophyll a and b, a + b content, and Rubisco activity. | [19] |
| Cotton (*Gossypium* sp.) | -Increase in ABA, proline, soluble sugar content, and antioxidant enzyme activities.<br>-Decrease in root size and biomass.<br>-Decrease in cotton yield and quality. | [20] |
| Wheat (*Triticum aestivum* L.) | -Decrease in photosynthetic pigments, chlorophyll a and b, and carotenoids.<br>-Increase in $H_2O_2$, loss of electrolyte, MDA, proline, and protein content. | [21] |
| Wheat (*Triticum aestivum* L.) | -Reduction in photosynthesis, stomatal conductance, relative water content (RWC), grain weight, and grain yield. | [22] |
| Wheat (*Triticum aestivum* L.) | -Decrease in length, fresh weight, and dry weight of the plant shoots.<br>-Decrease in membrane stability, chlorophyll a, b, relative water content (RWC), K, P, and N content in shoots.<br>- Increase in proline, sugar, and antioxidant enzyme activity (SOD, POD, CAT).<br>-Decrease in weight and number of grains per ear. | [23] |
| Canola (*Brassica napus* L.) | -Inhibition and decrease in germination percentage.<br>-Decrease in chlorophyll a and b content.<br>-Strong increase in $H_2O_2$ and MDA content, loss of electrolytes (EL).<br>- Increase in proline, antioxidant enzyme activity of SOD, POD, and CAT, concentrations of non-enzymatic antioxidants such as ascorbic acid (AsA), and glutathione (GSH). | [24] |
| Sugar beet (*Beta vulgaris* L.) | -Decrease in chlorophyll a, b, chl a + b carotenoids, and anthocyanins.<br>-Increase in proline content and synthesis, and accumulation of antioxidant activities (CAT, APX, POX) and radicals (ROS) that damage the cell membrane.<br>-Increase in MDA, $H_2O_2$, and membrane degradation, and decrease in membrane stability.<br>-Decrease in relative leaf water content (RWC) and root yield. | [25] |
| Tomato (*Solanum lycopersicum* L.) | -Decrease in height, shoot weight, chl a, b, a + b, and relative leaf water content (RWC).<br>-Increase in MDA and $H_2O_2$ content, proline accumulation, and SOD and APX enzyme activity. | [26] |
| Maize (*Zea mays* L.) | -Reduction in size and fresh and dry weight of shoots and roots.<br>-Decrease in chl a, b, a + b, and relative leaf water content (RWC).<br>-Decrease in net photosynthetic, transpiration rate, and stomatal conductance.<br>-Increase in ABA concentrations.<br>-Causes changes in the ultra-structure of chloroplasts and thylakoids. | [27] |

## 2.1. Effects of Water Stress on Germination

Drought can occur at any stage of plant growth. However, seed germination is one of the most sensitive and important stages of a plant's life cycle. Water stress has a profound effect on these early stages of development [5,28].

The first stage of germination is the uptake of water by the dried seeds. The process requires that the hydration threshold of the embryo is reached. This is an important prerequisite for the continued initiation of cell elongation and radicle development [28]. Successful seed germination depends on the availability of sufficient moisture for metabolic activation to break dormancy or convert stored food into edible forms [15,29]. Under water shortage conditions, seeds cannot absorb enough water, which slows germination and reduces the total number of plants per unit area. Several studies have shown that drought stress reduces germination and growth of seedlings including wheat (*Triticum durum* L.) [16], rice [30], ornamental sunflower (*Helianthus annuus* L.) [31], canola (*Brassica napus* L.) [24], maize, and sorghum (*Sorghum bicolor* L.) [32].

## 2.2. Mineral Nutrition

Mineral elements are essential for plant growth and differentiation in all phases of plant development. Reduced growth following water deficit is attributed to reduced nutrient uptake, acquisition, and redistribution [33,34]. Plants acquire nutrients based on soil water availability as nutrients diffuse and flow to the absorbing roots and are absorbed and transported into the plant due to water potential gradients and water fluxes between roots, xylem, and leaves [35]. Reduced nutrient uptake in drought-stressed plants may be due to decreased transpiration rates. In addition, when soil moisture is low, stomata may close, transpiration decreases, and, thus, water and nutrient flux decreases. With the reduction in nutrient uptake comes a reduction in the efficiency of nutrient translocation into plant tissues, resulting in stunted growth and slower biochemical processes [36–39].

Plant roots absorb nutrients and water from the soil based on their architecture. Under drought conditions, insufficient root functionality and slow water diffusion make roots inefficient in absorbing nutrients from the soil [22]. The integrity of the root membrane is essential for plant mineral nutrition. Therefore, maintaining membrane stability is an important aspect of a plant's drought tolerance. Damage induced to cell membranes under water stress conditions is an important factor in the disruption of plant ion homeostasis [37,38].

It is evident that nitrogen (N), phosphorus (P), potassium ($K^+$), and calcium ($Ca^{2+}$) are the major nutrients essential for plant growth and metabolism. Calcium plays an essential role in the structural and functional integrity of the membrane and other structures. Decreased $Ca^{2+}$ content in stress conditions has been observed in many plants. In maize, for example, $Ca^{2+}$ content in stressed leaves drops by about 50% while in roots it is higher compared to the control [40,41].

Soil N mineralization may be reduced by water stress, ultimately reducing N availability. Reduced nitrate uptake also affects biochemical activities such as nitrate reductase [42]. Under stress conditions, the activity in enzymes involved in food uptake decreases; for example, the nitrate reductase gene is significantly downregulated, and nitrate reductase activity is significantly reduced in plants such as durum wheat, rice, and common barley (*Hordeum vulgare* L.) [42,43].

Many plant species reported reduced $K^+$ levels during water shortage due to membrane damage and disruption of ion homeostasis. Plants with low $K^+$ levels are less resistant to water stress [41]. Potassium is also an important mineral element and plays a key role in water relations, osmotic adjustment, stomatal movements, and ultimately plant drought resistance. Many plant species have noted decreased $K^+$ content under water stress, mainly due to membrane alteration and disruption of ionic balance [37].

Low P levels result in the inhibition of many metabolic processes such as respiration, photosynthesis, cell division, and expansion. Water stress reduces P transfer from soil to root and its subsequent transport to the stem [43]. In dry soil, P mobility is reduced because

it moves primarily by diffusion. Therefore, the availability of sufficient water in the soil is an essential factor that improves P mobility and uptake [35,39].

### 2.3. Plant Morphology and Anatomy

Morphological and anatomical characteristics of plants are severely influenced by water stress [10,33]. Cell division and differentiation, followed by cell expansion, are basic conditions for plant growth, which are negatively affected by drought, thus damaging their yield [14]. Slower growth is accompanied by a loss of cell turgor pressure, which is an essential step in cell proliferation [29,44]. Under water stress, cell expansion slows or ceases, and plant growth is thus delayed. The decrease in water supply reduces the availability of essential nutrients to the plant, which therefore reduces growth, characterized by the decrease in various growth parameters such as plant height, fresh and dry weight of shoots and roots, number of leaves per plant, number of branches per plant, decreased leaf size, reduced number of stomata, thickening of leaf cell walls, cutinization of leaf surface, poorly developed conductive tissue, and induction of early leaf senescence [29,44].

Water stress also alters root structure and morphology. To increase water uptake under dehydration conditions, plants expand their roots and produce a branched root system [45]. Increased allocation of biomass to the roots under drought conditions and expansion of the plant root system generally lead to greater water uptake capacity [46]. Thus, under dehydrated conditions, the root to stem ratio generally increases, but the total plant biomass decreases [37,47]. Although water saving is an important result of reducing leaf area, it is the cause of yield decrease, as a result of decreased photosynthesis. The decrease in chlorophyll content is a typical symptom in the case of water stress that could modify the morphology of the plants. A high root to aerial part ratio under water stress conditions is related to the content of ABA and ethylene in roots and aerial parts and the interaction of these two hormones [48].

### 2.4. Photosynthesis

Photosynthesis is an important physiological process for plant growth. A reduction and/or inhibition of photosynthesis is one of the main effects of drought in plants [48]. Photosynthesis is particularly sensitive to the effects of water shortage. Soil drought and leaf water deficit lead to a progressive suppression of photosynthesis associated with functional and structural alterations and rearrangements of the photosynthetic apparatus [38,49].

The decrease in photosynthetic rate is the result of a reduction in stomatal conductance, which is one of the first strategies used by plants to decrease photosynthetic rate [34]. Carbon dioxide limitations due to prolonged stomatal closure lead to the accumulation of reduced photosynthetic electron chain compounds. The accumulation of these compounds can reduce molecular oxygen and give rise to the production of reactive oxygen species (ROS), thus causing oxidative damage in chloroplasts [33,44]. However, the decrease in photosynthesis under water stress can also be attributed to disturbances in biochemical processes. Exposure to stress can induce alterations in photobiological processes, resulting in damage to the photosystems I and II reaction centers (PSI and PSII), thereby compromising photosynthetic efficiency. Non-stomatal limitation of photosynthesis has been attributed to reduced carboxylation, reduced regeneration of ribu-lose-1,5-bisphosphate (PuBP), and reduced functional quantity of ribulose-bisphosphate carboxylase/oxygenase (RuBisCO) [3,4].

The key enzyme of carbon metabolism in the Calvin cycle is RuBisCO [14,33]. Its level in leaves is controlled by the rate of its biosynthesis and degradation. The amount and activity of RuBisCO decrease rapidly under water deficit conditions. The decrease in intra-cellular $CO_2$ slows down the Calvin cycle processes, resulting in less NADPH and ATP utilization. This reaction leads to a lack of regeneration of electron acceptors ($NADP^+$, $NAD^+$, and FAD), allowing electron transfer from the electron transport chain to oxygen, and increasing the production of reactive oxygen species (ROS) that cause oxidative stress [17,50].

There is a correlation between drought severity and decreased photosynthesis due to an increase in the activity of RuBisCO-binding inhibitors that reduce its activity. Water stress also disrupts cyclic and non-cyclic types of electron transport in light reactions of photosynthesis [8,33]. A lower rate of electron transport negatively affects the process of photophosphorylation (ATP biosynthesis) and the reduction in $NADPH/H^+$ [3,4]. These alterations cumulatively disrupt the photosynthetic apparatus under water stress conditions. Both PSI and PSII photosystems in chloroplasts are affected by water-deficient conditions mainly due to low electron concentration transport rate and accumulation of ROS following its biosynthesis [47].

In addition to RuBisCO, the activities of some other enzymes involved in carbon metabolisms, such as phosphoenolpyruvate carboxylase, NADP-malic enzyme, fructose-1, 6-bisphosphatase, NADP-glyceraldehyde phosphate dehydrogenase, phosphoribulokinase, sucrose phosphate synthase, and pyruvate orthophosphate dikinase, decrease linearly with decreasing leaf water potential under drought conditions [17,50]. Changes in photosynthetic pigments have also been detected in plants stressed by lack of water, showing reduced or even no pigmentation. Chlorophyll a and b decline in stressed plants, which directly affects plant biomass production. The reduction in photosynthetic pigments results in a reduction in energy consumption and carbon demand for chlorophyll synthesis [48]. According to reports in the literature, carotenoids are less sensitive to water stress than chlorophylls, which play an essential role in the antioxidant defense system under stressful conditions [46,51].

*2.5. Hormonal Balance*

Various plant hormones are synthesized in response to drought stress and manage processes related to drought tolerance mechanisms. Phytohormones, such as auxin, cytokinin, gibberellin, abscisic acid (ABA), ethylene, salicylic acid, and jasmonic acid, are routinely produced in the plant system [52,53]. These phytohormones act as chemical messengers in response to various abiotic stresses. After the perception of the stress signal, phytohormones are released and activate various physiological processes that are very important for growth and vegetative development. Water stress modifies the level of these phytohormones, resulting in the plant's ability to perceive and transduce the stress signal to regulate its gene expression [53,54].

Abscisic acid (ABA) is a stress hormone which accumulates in response to water stress to regulate plant–water balance and cellular tolerance to dehydration via gene induction and stomatal closure [35]. ABA is rapidly produced in plant chloroplasts and roots in response to water stress. ABA is considered a signaling molecule, which then moves to the plant shoots and leaves to trigger stomatal closure, resulting in decreased water loss from the leaf surface [44,54,55]. ABA reduces water loss through the regulation of stomatal movements and recovery of photosynthetic and chlorophyll fluorescence parameters. ABA also mediates root elongation to reach deep soil water under drought conditions [29,56].

Cytokinin (CK) is a phytohormone, which can promote stomatal opening, decrease root growth, and stimulate shoot growth. Water stress significantly reduces CK levels in plants in association with stomatal closure [54]. Reduced CK concentration facilitates adaptive plant responses by decreasing water loss and stimulating root growth, improving soil water exploration. CK modulates plant growth through changes in plant morphology and metabolism, acting as indirect factors limiting the damage caused by water stress, thus increasing plant survival under water-scarce conditions [56].

Drought stress significantly reduces auxin accumulation in plant tissues. The best-known auxin phytohormone is indole-3-acetic acid (IAA), a derivative of L-tryptophan excreted by plants [53]. IAA is known to modify the architecture of the plant root system by improving root length, total root surface area, and root branching that leads to improved water and nutrient uptake, as well as coordinates cellular defense against water stress [26]. When plants are exposed to drought and other stress conditions, various modulations of auxin synthesis, metabolism, transport, and activity take place, A decrease in the level of

IAA under stress conditions can increase the level of ABA in plants to induce modulation of growth by auxins [56].

Gibberellin (GA) is a plant hormone that acts as the primary plant growth regulator and promotes fruit ripening and seed germination. In addition to promoting plant growth, the hormone acts as a protector against stress. It can scavenge ROS and help plants during water stress, maintaining the photochemical efficiency of photosystem II [55,56].

Jasmonic acid (JA) is another key plant hormone that improves plant tolerance to water stress due to its important role in regulating stomatal closure. The rapid increase in endogenous JA content is an immediate plant response to water stress, but the content decreases to its baseline level with prolonged stress [56]. It plays an important role in drought tolerance by reducing oxidative damage and lipid peroxidation, stimulating ABA production and osmolyte accumulation [56,57]. Additionally, JA is able to affect enzymatic activities by modifying gene transcription, translation, and post-transcriptional modifications, as it can also adjust the water potential of plant cells [58].

In addition, ethylene (ET) has been reported to be a negative regulator of drought stress tolerance. Under stress, responses to ET lead to an overall effect on root growth, leaf expansion, photosynthesis, grain development, and leaf senescence. Reduction in shoot growth and the ethylene-mediated stomatal response to water stress are significantly dependent on ABA accumulation in shoots [37].

In addition to the accumulation of ABA, drought typically causes an increase in the accumulation of brassinosteroids, a hormone that promotes water retention in plants and reduces ion leakage in response to water scarcity and extreme temperatures [33,44,59]. Brassinosteroids have been implicated in a wide range of physiological responses in plants, such as stem elongation, pollen tube growth, leaf curvature, epinasty, ethylene biosynthesis, proton pump activation, vascular differentiation, regulation of gene expression, nucleic acid, protein synthesis, and photosynthesis [56].

Salicylic acid is a versatile stress-sensitive plant hormone that is significantly produced in plants to trigger induced systemic tolerance (IST) against biotic and abiotic stresses [10,14]. Salicylic acid may play a defensive role in drought stress resistance. Salicylic acid increases the activity of oxidative enzyme systems such as catalase (CAT) and superoxide dismutase SOD [13]. Endogenous salicylic acid content decreases significantly during moderate water stress treatment due to the increase in ABA content under the stress treatment. It has been proposed that in response to water stress, an antagonistic interaction between these two hormones occurs naturally in several species [26].

In general, plant hormones do not function in distinct pathways but rather depend on each other at different stages to control signaling pathways, development, and response to environmental stresses.

### 2.6. Relative Water Content (RWC) and Membrane Stability

Leaf relative water content (RWC) is an important indicator of plant water status that represents the balance between water uptake by root hairs and water loss through transpiration [5,60]. RWC, leaf water potential, stomatal resistance, transpiration rate, leaf temperature, and canopy temperature are important factors affecting plant water potential [43]. The decrease in RWC is one of the most common consequences of water stress. This decrease is proportional to the severity of drought. It also depends on the plant species, growth stage, and duration of stress [23,61].

Low RWC reduces the water potential of leaves and closes the stomata, which lower the transpiration rate, leading to the increase in leaf temperatures [62]. Elevated leaf temperatures can denature proteins, especially enzymes, and alter membrane stability, which causes an imbalance in cellular metabolites such as photosynthesis, respiration, and nutrient concentration, as well as the synthesis of essential macromolecules [63]. Membrane instability is mainly due to lipid peroxidation caused by reactive oxygen species produced by water stress [43,64].

*2.7. Oxidative Stress: Production of ROS*

Drought causes the overproduction and accumulation of ROS in plant cells such as superoxide radicals ($O^{2-}$), hydrogen peroxide ($H_2O_2$), singlet oxygen ($O_2$), and hydroxyl radicals (OH). ROS are produced by several cellular compartments including chloroplasts, mitochondria, and peroxisomes. Chloroplasts are a potentially important source because excitatory pigments in thylakoid membranes can mix with $O_2$ to form strong oxidants. ROS generation in peroxisomes is primarily due to increased photorespiration, which leads to $H_2O_2$ generation by glycolate oxidase [49,65]. In general, the increase in ROS production is proportional to the severity of drought [13,65]. It is associated with a variety of physiological and biochemical disorders resulting in oxidative damage to lipids, proteins, and other cellular macromolecules, including DNA and RNA, and can eventually lead to cell death [13].

Lipid peroxidation damages the plasma membrane and reduces its stability, leading to loss of cellular content and rapid dehydration. Malondialdehyde (MDA) is the end product of lipid peroxidation, and MDA content has been used as a marker of membrane lipid peroxidation [28]. Oxidative proteins can also be the result of oxidative stress, leading to loss of enzymatic activity and formation of cross-linked aggregates [38]. In addition, increased ROS accumulation promotes the degradation of chlorophyll molecules, ultimately reducing photosynthetic performance [66] (Figure 1).

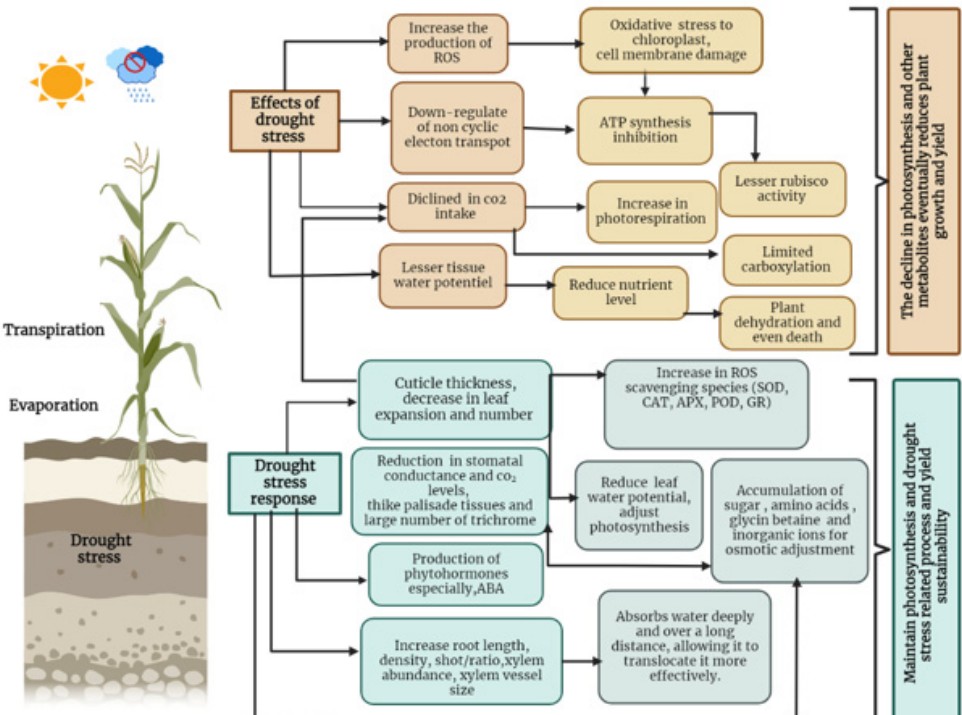

**Figure 1.** Effects of drought on plant growth and plant adaptation mechanisms.

## 3. Water Stress Tolerance Mechanisms Involved by Plants

Plants are able to sustain their growth to stay alive under water stress conditions. Tolerance is defined as the ability of plants to grow, thrive, and reproduce under severe stress conditions [67]. Drought tolerance is a complex defense mechanism in which plants use strategies to escape or avoid drought. However, these strategies are not exclusive and independent. Plant adaptation to water deficit is enabled by physiological, morphological, biochemical, and molecular responses. Responses can range from the molecular level to the whole plant level [68]. In addition, how plants respond to drought varies considerably with different levels of organization, growth stage, age, plant species, drought intensity, and duration [67,69].

### 3.1. Morphological Adaptation

Plants induce a series of structural changes to adapt to water deficit circumstances. Plants possess many adaptive traits to withstand this stress; among the most important is the architecture of the root system [70], which plays an important role in the response to water deficit. This architecture consists of the topology of the root system, the spatial distribution of main and lateral roots, and the length and number of various secondary roots. Roots have morphological plasticity that depends on the physical conditions of the soil [13]. In addition, plants will extend their root systems to increase water uptake and thus develop highly branched roots [71,72]. Thick and deep root systems are beneficial to the plant absorbing water deep and far [73].

In addition to the root system, water stress is also associated with changes in leaf structure. In response to drought, plants reduce water loss by closing their stomata. This reduces water loss through transpiration and improves water use efficiency [4,74]. The morphological changes in the leaf are not negligible and help the plant minimize water loss and better adapt to this stress. In addition, decreasing leaf area, increasing leaf thickness, and rolling leaves reduce transpiration. Cuticles and hairy leaves contribute to increased tissue water retention [4]. Lenticular wax biosynthesis on the surface of aerial plant parts is also closely related to plant resistance to water deficit [73].

Under water stress conditions, increasing the root to shoot ratio allows for water and nutrient uptake and maintenance of osmotic pressure. The root to stem ratio has been used as a criterion to define a plant's ability to withstand drought [67,71].

### 3.2. Osmolyte Accumulation: Osmotic Adjustment

Water stress affects the osmotic balance of plant cells. Osmotic adjustment is an important feature to maintain plant water status and delay tissue dehydration under water stress. Plants synthesize and accumulate osmolytes or compatible solutes that play an important role in plant responses to water stress. The primary function of compatible solutes is to prevent water loss and maintain cell turgor and the plant's water uptake gradient. The accumulation of these metabolites in the cells leads to an increase in osmotic potential (osmotic adjustment) and ultimately results in a greater capacity for water uptake by the roots and its retention in the cells [75,76]. In addition to their role in osmoregulation, compatible solutes show other functions such as protection of enzymes, membrane structure and integrity, maintenance of protein conformation at low water potential, neutralization of oxygen radicals, and stabilization of cellular macromolecule structures [76].

These compatible solutes are water soluble. They do not interfere with normal biochemical reactions, but replace water in the environment of nucleic acids, proteins, and membranes under water deficit conditions. The main categories of compatible solutes are: proline, glycine betaine, soluble sugars, polyols, and inorganic ions [48,77].

Proline is one of the best-studied and most accumulative amino acids. For example, under water stress, proline concentration can exceed 80% of the total amino acid pool in some plants, while it is only 5% in control plants [75,77]. The accumulation of proline in the cytoplasm indicates that the plant is under stress conditions, depending on the vegetative stage of the plants and the severity of the drought [48]. The increase in cytoplasmic concentration facilitates water uptake by the cells, maintains osmotic balance in the protoplasm, and reduces toxic ion levels [28]. Proline also protects cellular structures and binds to proteins to form a hydrophobic backbone that stabilizes macromolecules (DNA, chloroplasts, and mitochondria) against oxidative damage caused by ROS. Thus, proline stimulates the activities of antioxidant enzymes (CAT, SOD) that detoxify ROS [60]. The increase in proline level under drought conditions has been reported in different crops, including brown mustard (*Brassica juncea* L.) [78], chickpea (*Cicer arietinum* L.) [79], durum wheat [80], and sunflower [81].

Sugars provide carbon and energy for normal cellular metabolism and control plant growth and development. Drought stress causes rapid accumulation of soluble sugars such as glucose, sucrose, fructose, etc. Soluble sugars and alcohol sugars are key components

of osmotic adjustment in plants [25]. They are important osmo-protectants that provide membrane protection and remove toxic ROS generated during oxidative stress, due to their reducing ability that helps degrade ROS such as $H_2O_2$ [37,82].

Glycinebetaine (GB) (N,N,N-trimethylglycine) is another osmoregulatory molecule that plants produce in response to adverse growth conditions. It plays an important role in stress tolerance by stabilizing macromolecules through intermolecular water balance [83]. Its highly soluble physiological state and low viscosity due to its zwitterionic nature allow for cell protection through protein and membrane stabilization, PSII protection, pigmentation stabilization, and ROS reduction [84,85]. GB has been shown to preserve plasma membranes by reducing MDA levels [86].

Another strategy also acquires specific inorganic ions such as $Na^+$, $K^+$, $H^+$, $Ca^{2+}$, and $Mg^{2+}$, in addition to appropriate chemicals to maintain the intracellular water potential. These ions help alter the osmotic potential of the cell by regulating the concentration of inorganic ions and produce changes in cell structure and function [84,87]. Thus, the accumulation of inorganic ions allows plants to decrease energy consumption for osmotic adjustment by passively taking up more $Na^+$ [87].

### 3.3. Phytohormones

Phytohormones play an important role in plant growth and development as they respond to environmental signals. Plants modify their phytohormone levels to reduce the negative impacts of stress [88]. The variation in plant response to water stress is modulated and regulated by several plant hormones that promote phenotypic variation.

Hormones play important functions in the control of plant growth and development, as well as in the adaptation of plants to water shortage stress [56,89]. Under water deficit conditions, the endogenous levels of auxins, GA, and CK are reduced; on the contrary, ABA and ET levels are increased [29,38].

When plants are exposed to water stress, ABA alters root structure by decreasing ROS accumulation in the roots and also promotes root extension to increase deep water uptake [20,90]. Increased ABA also serves as a signal, initiating a signaling cascade in guard cells to regulate cell disruption [53,91]. ABA regulates osmotic homeostasis and acts as a chemical signal for the induction of genes related to enzymes that detoxify ROS, protein transporters, transcription factors, regulation of aquaporin activity, and post-transcriptional modifications that control guard cell swelling during stomatal closure [37,53]. It also promotes proline accumulation, dehydrating protein, and Late Embryogenesis Abundant LEA synthesis, which are involved in the regulation of osmosis and other plant defense systems [73,92].

Auxins such as IAA improve root structure, promoting new root growth by disrupting cytokinin-induced root apical dominance. A thick root system is necessary for drought tolerance. Auxins have an indirect but important effect. Water stress reduces endogenous auxin synthesis, which typically occurs when ABA and ethylene concentrations are increased [93]. Auxins also influence water stress tolerance by increasing metabolic balance and ROS detoxification and activating a variety of stress tolerance-related genes [93,94].

CKs are up- and downregulated during water stress. The increase or decrease in CK levels is determined by the duration and intensity of stress [94]. CKs play a role in plant adaptation to drought by controlling genes involved in $CO_2$ assimilation, electron transport rate, photosynthetic rate, and chlorophyll levels [95]. The decrease in CKs production and accumulation in roots is due to the mobilization of CKs to shoots, where they slow and prevent leaf and stem senescence. CK signaling interacts primarily with ABA to regulate plant yield under water stress [91]. ABA and CKs have opposite functions under drought conditions. Increased levels of ABA and reduced levels of CKs promote stomatal closure and minimize transpirational water loss [96].

### 3.4. Antioxidant Mechanisms: Reduction in ROS Generation

Drought stress induces the overproduction of ROS that cause oxidative damage to plants. However, ROS also act as a secondary message to activate plant defense mechanisms to ensure tolerance against this damage [85,97].

Under water scarcity conditions, plants develop defense mechanisms through the induction of enzymatic and non-enzymatic antioxidant activities. Antioxidant enzymes include CAT, peroxidase (POD), SOD, glutathione reductase (GR), ascorbate peroxidase (APX), glutathione peroxidase (GPX), monodehydro-ascorbate reductase (MDHAR), and dehydro-ascorbate reductase (DHAR) [40,98]. Non-enzymatic antioxidants include β-carotene, ascorbic acid, tocopherol, glutathione, flavonoids, phenolic compounds, and carotenoids [99,100].

Both mechanisms mitigate the effects of reactive oxygen species and help plants maintain cellular homeostasis and minimize oxidative damage [43,99]. Water stress increases the expression of genes, encoding antioxidant enzymes SOD, CAT, APX, and GPX, which are significantly influenced by species type, plant metabolic state, plant growth stage, and stress intensity and duration [45,101].

SOD is an important enzyme that acts as first line of defense and serves as a ROS scavenger. SOD converts $O^{2-}$ to hydrogen peroxide ($H_2O_2$) using different substrates to electron donors. CAT, POD, and APX break down $H_2O_2$ into molecular oxygen and water [65,102]. They also convert lipid hydroperoxide to alcohol, reducing lipid peroxidation and membrane damage [103]. Carotenoids are also involved in the elimination of harmful free radicals while allowing the protection of proteins of the photoreceptor complex, the stability of the thylakoid membrane, and the photo-oxidation of chlorophyll through their antioxidant properties. In addition, low-molecular-weight flavonoids have an excellent ability to scavenge free radicals and reduce lipid loss [49]. Antioxidants such as ascorbate and glutathione act as an enzyme cofactor as well as an electron donor/acceptor for APX and GPX [89,104].

An increase in enzymatic antioxidant activity under water deficit has been observed in several plant species including: rice [105], barley (*Hordeum vulgare* L.) [100], maize [106,107], wheat [23,101], and chickpea [108]. Therefore, both enzymatic and non-enzymatic defense mechanisms play essential roles in the detoxification and scavenging of ROS, as well as in increasing tolerance to drought stress (Figure 2).

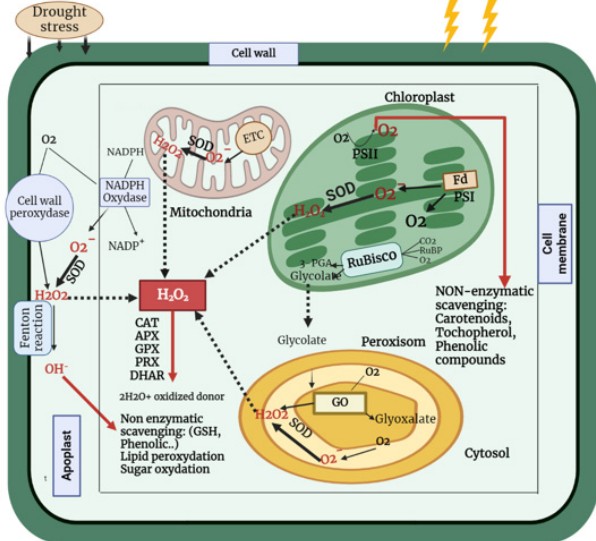

**Figure 2.** Scheme of ROS generation in various organelles during drought stress and their removal by enzymatic and non-enzymatic antioxidant mechanisms. Drought boosts ROS production in a variety of ways. During drought stress, low $CO_2$ levels in chloroplasts induce a reduction in $NADP^+$ regeneration via the Calvin cycle, resulting in an increase in electron leakage to $O_2$ to create $O^{2-}$ and, as a result, the production of more ROS. Furthermore, under a $CO_2$-limited environment, photorespiration in $C_3$

plant leaf peroxisomes results in the production of phosphoglycolate. It stimulates ribulose1 and 5 bisphosphate (RuBP) oxygenation, which might result in enhanced $H_2O_2$ generation, owing to phosphoglycolate breakdown. The rate of mitochondrial respiration rises as the severity of the drought grows, to compensate for the reduced rate of ATP synthesis in the chloroplast, resulting in greater ROS formation in the mitochondria. Scavenging of ROS occurs via enzymes such as ascorbate peroxidase (APX), catalase (CAT), peroxiredoxins (PRX), glutathione peroxidase (GPx), and dehydroascorbate reductase (DHAR), as well as non-enzymatic scavenging via ascorbic acid, glutathione, and flavonoids, among others.

*3.5. Molecular Responses: Stress Proteins, Transcription Factors, and Resistance Genes*

Plant responses to drought are extremely complex, as different morphological and physiological changes are mediated by complex signaling pathways that activate or inhibit downstream processes [109]. Along with physiological and biochemical changes, drought also induces changes at the molecular level regarding the induction/modification of gene expression, activation of regulatory proteins, and initiation of plant signaling to counter the effect of these stressors [110].

External drought stimuli are perceived and sensed by sensors which are located on the plasma membrane or exist freely in the cytosol. These extracellular signals are then transformed into intracellular signals via a variety of secondary messages present in the cell including ROS, ABA, sugars, and calcium $Ca^{2+}$ ions, as well as transcriptional regulators with important roles in various signaling pathways [84,111]. After detecting stress, downstream signaling activates several stress-responsive genes. There are two types of stress-responsive genes: (1) genes that encode protective proteins providing direct protection of the plant cell from stress such as heat shock proteins (HSPs) or chaperones, LEA proteins, osmoprotectants, ROS detoxifying enzymes, aquaporins (AQP), and transporters (sugars and proline), and (2) regulatory genes that enable stress signal perception and transduction within plant cells through general and specific regulatory-signaling pathways including transcription factors, mitogen-activated protein kinase (MAPK), and calcium-dependent protein kinase (CDPK) [109,111].

In addition, the expression of various stress-related genes can be controlled by transcription factors. TFs are able to bind to DNA sequences and activate and/or repress transcription of downstream target genes. These TF-controlled genes are further controlled by either the ABA-dependent or ABA-independent pathway [67,112].

ABA links stress perception to cellular transcriptional reprogramming for stress adaptation. In addition, protein kinase and phosphatase participate in protein phosphorylation and dephosphorylation, respectively. Protein phosphorylation and dephosphorylation play an important role in plant tolerance to abiotic stress [113]. Several protein kinases are reported to be expressed in response to osmotic stress. Mitogen-activated protein kinase (MAPK) cascades are one of the major cell signaling components in eukaryotic cells and are generally activated in response to hyperosmotic stress. There is considerable evidence that a very early step in water deficit signal transduction in plant cells involves MAPK pathways. For example, a change in plasma membrane fluidity could induce a conformational change in a receptor that activates a downstream kinase cascade. We also find that sucrose-related kinases without fermentation (SNF1) are induced in response to hyperosmotic stresses. CDPKs are another important group of kinases that are activated in response to osmotic stress and activate calcium signaling [84,112].

Thus, stress proteins (AQP, LEAs, and HSPs) are highly expressed during abiotic stress. They are mostly water soluble and provide stress tolerance for plants by hydrating cellular structures [110,114]. LEA proteins are the most abundant stress proteins and are related to both water and cold stress in plants. These proteins are active in seeds that contain high levels of ABA [115]. These proteins have a low molecular weight (10–30 kDa) and they accumulate at the time as seed desiccation (at a later stage of embryogenesis), as well as during persistent water deficit conditions, under cold conditions, or high salinity [110,116].

LEA are expected to play an important role in ion sequestration during cellular dehydration or water deficit. LEA proteins are hydrophilic and thermostable. Their structural flexibility facilitates interactions with other macromolecules, such as membrane proteins, leading to cell membrane stability during water stress [114,116]. According to research, LEA proteins function as stabilizers, hydration buffers, membrane protectors, antioxidants, ROS scavengers, ion chelators, and chaperones [117,118]. These proteins were first detected in the late development of cotton (*Gossypium hirs spit*) seeds by Dure in 1981. The LEA gene family has later been detected in many crops including: cotton (*Gossypium* spp.) [119], wheat [115], and tomato (*Solanum lycopersicum* L.) [120].

AQP are water transport channels in plants. They are highly selective in regulating processes such as seed germination and stomatal movements [121,122]. They are intrinsic membrane proteins that simplify and control the passive exchange of water molecules across plant membranes. In plants, they are widely found in plasma and vacuolar membranes. They are abundantly expressed in roots, where they are involved in soil water uptake [122]. It is widely accepted that in most plant species, water uptake and transcellular water flow in roots are largely mediated by PIP (Plasma membrane Intrinsic Protein) and TIP (Tonoplast Intrinsic Protein) [123]. These are the most abundant AQP in the plasma membrane and tonoplast of plant cells, respectively [124,125]. During water stress, aquaporin regulates root hydraulic conductivity, and when moisture is low, the expression of aquaporin-encoding genes is downregulated, reducing the ability of roots to take up water. They maintain cellular homeostasis by limiting water loss and increasing membrane permeability [111].

HSPs are stress proteins that stabilize proteins from denaturation or activate their degradation after plants have overcome a period of stress [126]. HSP production is not limited to high temperature stress. It has been reported that some HSP genes are also expressed under osmotic stress in several plants: rice [127], maize [128], chickpea [91], *Arabidopsis thaliana* [126], wheat [129], and quinoa (*Chenopodium quinoa*) [130]. For example, HaHsp17.6 and HaHsp17.9 genes are expressed in sunflower under water stress; homologs to these genes are found in the plant *Craterostigma plantagineum,* and their level increases under drought stress via ABA-dependent mechanisms [110].

HSPs help improve various physiological phenomena, including membrane stability, water and nutrient efficiency, and photosynthesis [128]. Based on previous research, HSPs can help plant defense through three roles: (1) refolding denatured proteins, (2) participating in the process of refinement of newly produced proteins, and (3) eliminating protein aggregation and, consequently, cellular homeostasis [114,131].

## 4. Osmotolerance through Plant Growth-Promoting Rhizobacteria

Drought tolerant PGPR are considered an effective alternative for agriculture under water stress. The use of PGPR is considered a synergistic long-term biological technique to cope with the adverse effects of drought. Through their colonization potential and symbiotic interactions with plants either on the rhizosphere surface or within plant cells, microbial communities are currently receiving increased attention for improving crop productivity and providing resistance to abiotic stress [131–134]. Several beneficial drought-resistant rhizobacteria are being isolated and tested to induce water stress tolerance in plants under controlled conditions. PGPR such as *Paenibacillus polymyxa*, *Achromobacter* sp., *Azospirillum brasilense*, *Pseudomonas* sp., *Burkholderia*, *Arthrobacter*, and *Bacillus* have been shown by several studies to improve drought tolerance in *A. thaliana* [135], tomato [26,136], wheat [64,88], and maize [2]. PGPR have remarkable potential to modulate the physiological response against water scarcity, thereby ensuring plant survival (Table 2).

**Table 2.** Alleviation of drought stress in several crops through the application of new and diversified plant growth-promoting rhizobacteria.

| PGPR Strains | Plant Species | Mechanisms | Beneficial Features Related to Drought Tolerance | Reference |
|---|---|---|---|---|
| *Pseudomonas fluorescens; P. putida* | Myrtle (*Myrtus communis* L.) | IAA ,Nitrogen fixation | -Enhancement of plant biomass (shoot and root trimming), greater water and nutrient absorption. -Increase in chlorophyll a, b, and carotenoid concentration while decreasing electrolyte loss and MDA. -Stimulation of enzymatic (CAT, SOD, POD) and non-enzymatic (Phenol and flavonoids) antioxidant defense. | [137] |
| *Achromobacter xylosoxidans; Enterobacter cloacae* | Maize (*Zea mays* L.) | ACC deaminase IAA, | -Increase in plant size, height, shoot dry weight, grain production, photosynthetic rate, transpiration rate, chlorophyll a, b, and a + b, and stomatal conductance. | [138] |
| *Bacillus* sp.; *Azospirillum lipoferum; Azospirillum brasilense* | Wheat (*Triticum turgidum*. L) | IAA, ABA, CK | -Increase in seed germination and root system growth. -Improved plant biomass (fresh and dry weight), RWC, photosynthetic pigment concentration (Chl a, b, and carotenoids), nutrient absorption, and plant yield. -Decrease in EL while maintaining membrane stability. -Increase the amount of osmolytes (proline and sugar) and antioxidant enzymes (CAT, SOD, APX). | [5] |
| *Bacillus amyloliquefaciens* 54 | Tomato (*Solanum lycopersicum*.L) | EPS | -Improved RWC, root vigor, antioxidant enzyme activity (SOD, APX, CAT, POD). -Decrease in MDA content and increase in nced1 gene transcription and ABA content. | [136] |
| *Azospirillum lipoferum* | Wheat (*Triticum aestivum*. L) | - | -Decrease in oxidative membrane damage. Improved plant growth, RWC, grain yield, and leaf photosynthesis. -Increase in antioxidant enzymes such as SOD, POD, and non-enzymatic phenolics. | [139] |
| *Azospirillum baldaniorum* Sp245 | Basilic (*Ocimum basilicum* L.) | ISR | -Increase in the content of phytohormones in plant xylem (SA, JA, ABA, etc.) -Increase in chlorophyll a, b, a + b, and anthocyanin concentration, as well as photosynthesis. | [140] |
| *Streptomyces pactum* Act12 | Wheat (*Triticum aestivum* L.) | - | -Increase in root and shoot growth and fresh weight, as well as osmolyte sugar and proline accumulation, RWC. -Decrease in MDA content and increase in ABA content and non-enzymatic antioxidant. -Stress resistance gene expression SnRK2, EXPA2, EXPA6, P5CS | [141] |
| *Bacillus* sp. WM13-24; *Pseudomonas* sp. M30-35 | Ray-grass (*Lolium perenne* L.) | IAA, ACC deaminase, EPS | -Improved root architecture (RSA), shoot trimming, and fresh and dry pea shoots and roots -Increase in photosynthetic rate, RWC, total chlorophyll, total nitrogen, and phosphorus levels in shoots. -Decrease in MDA content, relative membrane permeability (RMP), and $H_2O_2$ buildup. -Increase in antioxidant enzyme activity (CAT, POD) and proline accumulation. | [142] |
| *Pseudomonas fluorescens* WCS417; *Bacillus amyloliquefaciens* GB03 | Peppermint (*Mentha piperita* L.) | IAA, ACC deaminase | -Increase in growth through increasing leaf area, fresh and dry weight of shoots and roots. -Decrease in MDA levels and increase in both enzymatic (PX, SOD) and non-enzymatic (Phenol, ascorbic acid) antioxidant activities. | [45] |
| *Pseudomonas azotoformans* FAP5 | Wheat (*Triticum aestivum* L.) | EPS ,ACC deaminase, IAA | Increase in the rate of germination. Increase in plant biomass (root and leaf size, dried and fresh root and leaf weight). -Increase in RAS/RT ratio, soil aggregation, photosynthesis, and chlorophyll content. -Reduction in MDA levels and antioxidant enzyme activity. | [64] |
| *Bacillus subtilis* | Wheat (*Triticum aestivum* L.) | ACC deaminase | -Improved plant development in terms of size, fresh peas, and dry weight of shoots and roots. -Increase in RWC, chlorophyll levels, minerals, and antioxidant enzyme activity (SOD, CAT, POD) | [143] |

**Table 2.** *Cont.*

| PGPR Strains | Plant Species | Mechanisms | Beneficial Features Related to Drought Tolerance | Reference |
|---|---|---|---|---|
| *Bacillus* spp. | Rice (*Oryza sativa* L.) | ACC deaminase, IAA, EPS. | -Improved germination, plant biomass, and yield while reducing membrane and oxidative damage. | [144] |
| *Bacillus* sp. MN-54; *Enterobacter* sp. FD-17; *Pseudomonas fluorescens* 1 | Maize (*Zea mays* L.) | ACC deaminase ,ABA | -Increase in RWC, chlorophyll a, b, and carotenoid concentration, and photosynthesis. Increase in the activity of antioxidant enzymes (SOD, GPX, CAT). | [145] |
| *Planomicrobium chinense* P1; *Bacillus cereus* P2 | Sunflower (*Helianthus annuus* L.) | Phosphate solubilization Production of phytohormone (GA, AIA) | -Improved shoot length, root length, fresh and dried root biomass. -Increase in quantity of chlorophyll, carotenoids, leaf proteins, sugar, and phenolic substances. -Reduction in MDA levels. | [146] |
| *Bacillus subtilis*; *Azospirillum brasilense* | Wheat (*Triticum aestivum* L.) | EPS | -Increase in germination rate, seedling vigor index, and promptitude index. -Improved plant biomass (height, dry weight, fresh weight of shoots and roots, leaf area). -Increase in water potential and chlorophyll content (Chl a, b, and carotenoids). -Increase in osmolyte content (proline, amino acids, sugars and proteins) and antioxidant enzymes (SOD, CAT, and POD). | [69] |
| *Pseudomonas* sp. N66 | *Sorghum bicolor* L. | ACC deaminase | -Improved growth and root system architecture (RSA) via regulation of phytohormones (GA, AIA, CK). -Activates systemic tolerance induced (ISR) by the signaling hormones brassinolides, SA, and JA and the signaling molecules sphingosine and psychosine. -Reduced ethylene levels. -Increase in antioxidant activity and osmolyte accumulation (proline, glutamic acid, choline). | [68] |
| *Bacillus subtilis* HAS31 | Potato (*Solanum tuberosum* L.) | ACC deaminase | -Increase in chlorophyll concentration, soluble proteins, total soluble carbohydrates, and antioxidant enzyme activity (CAT, POD, SOD). | [147] |
| *Pseudomonas fluorescens* | Maize (*Zea mays* L.) | ACC deaminase | -Enhanced chl a, b, and total chl content, as well as F0 and Fm photosynthetic activities and yield. -Increase in proline accumulation, total soluble sugars, and antioxidant activity (CAT, POD). | [148] |
| *Pantoea alhagi* LTYR-11ZT | Wheat (*Triticum aestivum* L.) | IAA, EPS | -Improved growth and fresh weight of shoots and roots. -Increase in the amount of sugar and chlorophyll and reduction in MDA content. | [149] |
| *Bacillus* sp.; *Pseudomonas* sp. | Tomato (*Solanum lycopersicum* L.) | ACC deaminase, EPS, IAA. | -Increase in the rate of germination. -Increase in plant biomass and photosynthesis. -Improved resilience to drought and recovery. | [150] |
| *Ochrobactrum anthropi* DPB13; *Pseudomonas palleroniana* DPB15 *Pseudomonas fluorescens* DPB16; *Pseudomonas palleroniana* | Finger millet (*Eleusine coracana* L.) | ACC deaminase, IAA. | -Increase in growth metrics (shoot and root size and weight), total chlorophyll, and nutrient supply (N, P, K Na$^+$, Ca$^{2+}$). -Reduction in $H_2O_2$ and MDA levels and improved membrane stability. -Increase in proline, phenol, and antioxidant enzyme activity accumulation (SOD, GPX, APX). | [151] |
| *Pseudomonas* sp.; *Serratia marcescens* | Wheat (*Triticum aestivum* L.) | ACC deaminase, IAA, EPS | -Decrease in reactive oxygen species. -Enhanced osmolyte accumulation, chlorophyll, and carotenoid content in plant leaves, and Zn and Fe content in leaves are all benefits of improved plant water status. | [152] |
| *Bacillus cereus* AKAD A1-1, *Pseudomonas otitidis* AKAD A1-2; *Pseudomonas* sp. AKAD A1-16 | Soybean (*Glycine max* L.) | ACC deaminase, IAA, EPS | -Improved plant biomass (height, fresh weight, dried shoot and root weight), RWC, and chlorophyll content. -Increase in proline, soluble carbohydrates, and protein content. -Decrease in MDA and $H_2O_2$ levels. | [153] |

**Table 2.** *Cont.*

| PGPR Strains | Plant Species | Mechanisms | Beneficial Features Related to Drought Tolerance | Reference |
|---|---|---|---|---|
| *Ochrobactrum pseudogrignonense* RJ12; *Pseudomonas* sp. RJ15; *Bacillus subtilis* RJ46 | Pea (*Pea sativum* L.) Black gam (*Vigna mungo* L.) | ACC deaminase. | -Improved plant biomass, root and shoot length, and seed germination. <br> -Increase in the accumulation of osmolytes (proline, phenol), chlorophyll content, and RWC. <br> -Decrease in ACC accumulation in plants and decrease in production of the ACC oxidase gene. <br> -Increase in antioxidant enzyme activity (CAT, POD). | [154] |
| *Pseudomonas lini* DT6; *Serratia plymuthica* DT8 | Jujubier (*Ziziphus jujuba*) | ACC deaminase, IAA EPS | -Increase in plant biomass via pruning (fresh and dry peas from branches and roots), RWC. <br> -Increase in the phosphate and nitrogen content of the leaves. <br> -Reduction in the amount of MDA and ABA in the leaves. <br> -Increase in SOD and POD antioxidant enzymes. | [155] |
| *Bacillus amyloliquefaciens* MMR0 | Candle millet (*Pennisetum glaucum*) | ACC deaminase | -Increase in seedling vigor and germination rate. <br> -Enhanced plant growth factors such as total chlorophyll and RWC. <br> -Reduction in MDA content, whereas proline, APX, and SOD levels were elevated. <br> -Significant reduction in the relative expression of drought response marker genes (DREB-1E) and ethylene response factor (ERF-1B), with enhanced APX1 and SOD1 gene expression. | [156] |
| *Pseudomonas* SP2 | *Arabidopsis thaliana* | ACC deaminase, ABA ,IAA | -Improved plant biomass, RWC, and chlorophyll levels and reduction in MDA. <br> -Improved SP2 colonization and survival. <br> -Increase in osmolyte, glycine-betaine, and proline accumulation. | [157] |
| *Bacillus subtilis* | Okra (*Abelmoschus esculentus* L.) | - | -Increase in osmolyte (sugar and proline) accumulation, non-enzymatic antioxidants (glutathione GSH, ascorbate AsA), and enzymatic antioxidants (SOD, CAT, APX, GR, DHAR, MDHAR). <br> -Increase in photosynthesis and reduction in MDA and $O^{2-}$ levels. | [158] |
| *Bacillus altitudinis* FD48; *Bacillus methylotrophicus* RABA6 | Rice (*Oryza sativa* L.) | IAA, CK, GA, ACC deaminase | -Increase in root and shoot growth, root to shoot ratio R/S, RWC, photosynthetic pigments (Chl a and b), and proline content. <br> -Reduction in oxidative stress by increasing the activity of antioxidant enzymes (CAT, SOD, POD, APX). <br> -Increase in the number of productive tillers, the quantity of seeds, the grain weight, and the harvest index. | [159] |
| *Bacillus thuringiensis* Rhizo SF 23; *Bacillus subtilis* Rhizo SF 48 | Sunflower (*Helianthus annuus*) | ACC deaminase. | -Improved plant growth, proline content, and antioxidant enzyme activity (APX, SOD). <br> -Reduction in MDA content. | [160] |
| *Klebsiella* sp.; *Enterobacter ludwigii*; *Flavobacterium* sp | Wheat (Triticum aestivum L.) | EPS, ACC deaminase, IAA. | -Increase in the quantity, size, and dry weight of roots and shoots, as well as RWC and RAS/RT. <br> -Reduction in $H_2O_2$, MDA, and EL while preserving membrane integrity and stability. <br> -Decrease in proline and soluble sugars accumulation. <br> -Enhanced expression of resistance genes (DREB2A, CAT1). | [161] |
| *Arthrobacter arilaitensis*; *Streptomyces pseudovenezuelae* | Maize (*Zea mays* L.) | ACC deaminase, phosphate solubilization, IAA, siderophore production and ammonia | -Increase in leaf number, height, fresh weight, dry weight of shoots and roots, and chlorophyll content. | [162] |
| *Bacillus* spp. | Maize (*Zea mays* L.) | IAA, nitrogen fixation, Phosphate solubilization | -Enhanced growth parameters (height, fresh weight, dry weight). <br> -Increase in the amount of photosynthetic pigments (chl a, b, and total chl). <br> -Decrease in MDA, $H_2O_2$, and antioxidant enzyme activity (SOD, APX, POD, CAT). | [11] |

| PGPR Strains | Plant Species | Mechanisms | Beneficial Features Related to Drought Tolerance | Reference |
|---|---|---|---|---|
| *Klebsiella* sp. (LEW16) | Wheat (*Triticum aestivum* L.) | EPS, IAA | -Increase in germination rate, root morphology, and seedling growth parameters (height, fresh and dry weight, and root diameter). | [163] |
| *Bacillus amyloliquefaciens* QST713 | Alfalfa (*Medicago sativa* L.) | IAA, EPS | -Increase in plant growth and biomass (dry, fresh weight of shoots and roots), RWC, photosynthesis, chlorophyll a, b, carotenoid content, and antioxidant enzyme activities (SOD, CAT, POD, APX).<br>-Reduction in $H_2O_2$, $O^{2-}$ content.<br>-Induction of systemic resistance. | [164] |
| *Gluconacetobacter diazotrophicus* Pal5 | Rice (*Oryza sativa* L.) | Nitrogen fixation, IAA | -Improved plant development (leaf area, root and shoot size), root system architecture (RSA), and RWC.<br>-Increase in photosynthetic activity (Fv/Fm), photosynthetic pigment (Chll a, b, a + b, and carotenoids), phytohormone concentration (AIA, GA1, GA3, and zeatin), and osmolyte accumulation (trehalose, -tocopherol).<br>-Activation of the expression of genes involved in root formation. | [165] |
| *B. amyloliquefaciens* RHF6; *B. amyloliquefaciens* LMG9814; *Bacillus* sp. AGS84 | Spinach (*Spinacia oleracea* L.) | IAA, phosphate solubilization, $NH_3$ | -Increase in plant biomass and germination rate.<br>-Increase in the content of photosynthetic pigments (Chll a, b, a + b, and carotenoids), as well as Fv/Fm photosynthesis. | [166] |
| *Planomicrobium chinense* strain P1; *Bacillus cereus* strain P2 | Wheat (*Triticum aestivum* L.) | EPS, IAA | -Increase in plant biomass (fresh weight, dry weight of roots and shoots), RWC, and micro/macronutrient accumulation (such as Ca, Mg, Na, K, Cu, Cr, Zn, and Fe).<br>-Decrease in MDA content, antioxidant enzyme activity (POX, CAT, POD), and proline content. | [167] |
| *Mesorhizobium cicero*.; *B. subtilis*; *B. mojavenss* | Chickpea (*Cicer arietinum* L.) | IAA, ACC deaminase | -Improved plant growth (height, fresh peas, shoot and root dry weight), nutrient accumulation in seeds and leaves, root colonization, and survival.<br>-Improvement in seed yield. | [168] |
| *Ochrobactrum* sp. EB-165; *Microbacterium* sp. EB-65; *Enterobacter* sp. EB-14; *Enterobacter cloacae* strain EB-48 | Sorghum (*Sorghum bicolor*.L.) | IAA, N2 fixation, phosphate solubilization, siderophore synthesis, ACC deaminase | -Increase in proline accumulation, relative water content (RWC), and membrane stability index.<br>-Improved root system development and topology (height, fresh weight and dry weight).<br>-Induction of particular gene upregulation: sbP5CS2, sbP5CS1. | [169] |
| *Bacillus endophyticus* PB3; *Bacillus altitudinis* PB46; *Bacillus megaterium* PB50 | Rice (*Oryza sativa* L.) | ACC deaminase, IAA, EPS, GA | -Increase in (RWC), total sugars, proteins, proline, phenolics, potassium, calcium, and ABA.<br>-Increased the expression of stress-related genes (LEA, RAB16B, HSP70, SNAC1, and bZIP23). | [170] |
| *Pseudomona helmanticensis*; *Pseudomonas* | Wheat (*Triticum aestivum* L.) | Phosphate solubilization, IAA. | -Increase in seed and plant growth (height, fresh weight, and dry weight of shoots and roots).<br>-Increase in the availability of phosphorus in the soil as well as its uptake by the shoots. | [171] |
| *Streptomyces* sp. RA04; Nocardiopsis sp RA07 | Sorghum (*Sorghum bicolor* L.) | ACC deaminas, Phosphate solubilization, IAA, sederophore synthesis | -Increase in plant growth, photosynthesis, and chlorophyll levels.<br>-Reduction in MDA level while increasing the activity of antioxidant enzymes SOD, CAT, and APX. | [172] |
| *Streptomyces* strains IT25; *Streptomyces* C-2012 | Tomato (*Solanum lycopersicum* L.) | ACC deaminas, Phosphate solubilization, sederophore synthesis | -Increase in plant growth and fruit output.<br>-Reduction in CAT and GPX activity while increasing relative water, proline, and sugar content.<br>-Inhibition of ERF1 and WRKY70 stress transcript genes. | [173] |
| *Bacillus megaterium*; *B. licheniformis* | Wheat (*Triticum turgidum* L.) | ACC deaminase IAA, | -Increase in seed germination.<br>-Increase in plant development (shoot and root), RWC, chlorophyll a, b, and carotenoids, and osmolyte accumulation (proline, sugar, protein).<br>-Decrease in MDA and electrolytes (EL).<br>-Increase in the activity of antioxidant enzymes (SOD, APX, PODL, CAT, GR).<br>-Stimulation of water stress-related protein production. | [174] |

PGPR improve plant tolerance to drought through various direct and indirect mechanisms including changes in root system morphology and architecture, production of phytohormones, extracellular polysaccharides (EPS), ACC (1-aminocyclopropane-1-carboxylate) deaminase, volatile chemicals, and osmolyte accumulation. They may also enhance the antioxidant defense system and induce transcriptional regulation of stress response genes [1,44] (Table 2).

*4.1. Improvements in Physiological and Morphological Processes*

4.1.1. Germination Enhancement

Seed germination is a critical stage in the plant life cycle. The individual performance of a plant during the early stages of its life can have a significant impact on its development and yield. The use of PGPR to increase the rate of seed germination has been proven by several research studies. A higher germination rate indicates better growth and development of seedlings, which is essential for agricultural yields [64,175–177].

Seed treatment with the osmotolerant bacterial strain *Enterobacter* spp. improved the germination rate and seedling vigor index of both tomato Arka Meghali and Pusa Ruby varieties under osmotic stress conditions compared to untreated seeds [178]. Inoculation of onion (*Allium cepa* L.) seeds under osmotic pressure conditions (-0.8 MPa) by an osmotolerant actinobacterium *Citricoccus zhacaiensis* B-4 resulted in an improved germination rate and seedling vigor [179]. The increase in germination rate of chickpea in the presence of two concentrations of PEG (15% and 30%) was proved through inoculation of *P. putida* (RA) in two varieties of chickpea compared with untreated seeds [176]. A study by Galeano et al. [180] found that maize seed inoculation with *B. cereus* VBE23 under water stress significantly increased germination rate and seed germination parameters. Inoculation of *Burkholderia* sp. L2 and *Bacillus* sp. A30 also proved an increase in germination of tomato seeds under water stress [181]. A significant increase in germination of wheat and cucumber (*Cucumis sativus* L.) seeds occurred under osmotic stress when inoculated with *Paenibacillus beijingensis* BJ-18 and *Bacillus* sp. L-56 [182].

4.1.2. PGPR Improve Root Architecture

The root system is most affected by drought due to direct exposure with the soil. Modification of root system architecture (RSA) is an important response of plants to water deficits [183,184]. Several studies have shown that inoculation of plants under stressful conditions with osmotolerant PGPR strains improves plant growth and RSA. The development of roots and the formation of absorptive hairs lead to an increase in the soil surface occupied by the roots and consequently confer plants with better absorption [185,186]. PGPR can modify the root architecture through the secretion of low-molecular-weight signaling molecules such as 2,4-diacetylphloroglucinol (DAPG), produced by fluorescent *Pseudomonas* [187].

A study by Jochum et al. [188] demonstrated that *Bacillus* sp. 12D6 and *Enterobacter* sp. 16i inoculation of wheat (*Triticum aestivum* L.) and maize under water stress conditions resulted in increased root size, diameter, root branching, greater surface area, and greater number of root branches associated with improved water stress tolerance and total crop yield when compared to the non-inoculated plant. Inoculation of wheat with the endophytic strain *B. subtilis* improved the size and fresh and dry weights of roots and shoots under water deficit conditions [143]. Inoculation of *P. putida* MTCC5279 of chickpea plants improved the size of primary roots and increased the number of lateral roots of absorbing hairs. Additionally, it allows an increase in root nodules for better nitrogen fixation under water stress conditions [176].

Thus, inoculation with *Ochrobactrum* spp. strain NBRISH6 improved root length, dry weight, and hair branching of maize under stress conditions [189]. Inoculation of *Bacillus* spp. into sorghum under drought stress resulted in improved soil moisture and increased root dry biomass, thus improved sorghum seedling growth [184]. *Bacillus subtilis* GOT9 improve drought and salinity resistance in *A. thaliana* and cabbage (*Brassica campestris* L.) by increasing lateral root growth and development [190].

### 4.1.3. PGPR Improve Shoot Growth

Improved shoot growth under drought conditions through the application of PGPR is useful for increasing crop growth and productivity. Plants inoculated with PGPR strains have proven near normal shoot growth rates, resulting in increased crop productivity [191]. It has been shown in several studies that PGPR improved biomass and yield of plants under water stress. Naveed et al. [192] and Naveed et al. [193] proved that under drought stress, maize and wheat plants inoculated with the endophytic bacteria *Enterobacter* sp. FD17 and *B. phytofirmans* strain PsJN had more root and shoot biomass than non-inoculated plants. García et al. [194] also showed that inoculation of maize seedlings with osmotic stress tolerant *Azospirillum* sp. bacteria promoted plant growth when water is limited. Inoculation of *Planomicrobium chinense* P1 and *B. cereus* into sunflower significantly increased shoot and root length, fresh and dry shoot, and root biomass by providing resistance during drought [146]. Inoculation of *P. putida* NBRIRA and *B. amyloliquefaciens* NBRISN13 alone or in consortium improved growth parameters and yield of chickpea grown under water deficit conditions [195].

### 4.2. PGPR Improve Relative Water Content RWC

It has been shown by several studies that plants inoculated with PGPR have higher RWC compared to non-inoculated plants. The increase in RWC improves the drought resistance of plants. The mechanism by which PGPRs improve plant RWC is not yet clear. Researchers speculate that PGPR may help increase plant RWC through exogenous synthesis of osmoprotectants, such as proline, that increase cell membrane stability and decrease electrolyte leakage and water loss. PGPR can also help plants through the production of hormones such as IAA that enhance root growth and lateral root formation, thereby increasing water uptake, decreasing leaf transpiration, improving nutrition and physiology, and controlling stomatal closure and metabolic activities. [196,197].

A study showed that inoculation of wheat and *A. thaliana* with *A. brasilense* Sp245 increased water potential and relative water content RWC [197]. Pistachieri (*Pistacia vera* L.) plants inoculated with *Staphylococcus sciuri* had a 10% increase in RWC compared to non-inoculated seedlings at the same combined salinity and drought stress levels compared to non-inoculated plants [198]. Sandhya et al. [199] showed improvements in RWC in maize after inoculation with *P. putida* GAP-P45 by increasing proline content compared to non-inoculated plants. Inoculation of harefoot clover (*Trifolium arvense* L.) with the IAA-producing endophyte *P. azotoformans* ASS1 improved leaf RWC in the absence and presence of drought stress [200].

### 4.3. Osmotic Adjustment through the Synthesis and Accumulation of Solute Compatible

Osmotic adjustment is one of the key adaptations at the cellular level that helps plants tolerate drought-induced damage. PGPR inoculation can affect osmoregulatory capacity by increasing soluble sugar, protein, and proline contents, leading to a higher water potential gradient and thus improving plant water uptake and growth under stress conditions [174,201]. The most commonly acclimated osmolytes in water-stressed plants and bacteria include proline, trehalose, and glycine betaine [202,203].

In general, these drought-resistant rhizobacteria, such as *A. brasilense*, *Bacillus* spp. and *B. thuringiensis* excrete proline, glycine betaine, trehalose, and soluble sugars in response to water stress [204], which function synergistically with plant-derived osmolytes and promote plant growth [1]. It was observed that plants with increased levels of free amino acids and soluble sugars after treatment with PGPRs tolerate drought stress. [14,131].

Proline has been recognized as a multifunctional signal molecule, which at high concentration contributes to increasing stress tolerance in plants [48]. Thus, a positive correlation has been established between proline accumulation and drought tolerance in plants [12]. It protects plants from stress through various means, such as detoxification of reactive oxygen species, contribution to osmotic adjustment, and stabilization of native enzyme, protein, and membrane structures [205]. *Sphingomonas* sp. LK11 improved drought

tolerance in inoculated soybean (*Glycine max* L.) plants by increasing the synthesis of sugars and amino acids (glycine, glutamate, and proline) [206]. In another study, inoculation of a drought-tolerant bacterium, *B. subtilis*, into drought-stressed okra (*Abelmoschus esculentus* L.) was shown to increase appropriate solutes such as total sugar and proline [158]. The increased accumulation of proline in inoculated plants contributes to increased drought tolerance by supporting the osmotic potential of drought-exposed plants, through activation of the gene of proline synthesis [158]. Li. et al. [141] showed that *S. pactum* Act12 enhances the development of PEG-induced drought-stressed wheat plants by increasing proline and sugar levels in inoculated plants. Sugar accumulation under water stress preserves cell turgor; in addition to functioning as an osmolyte, sugar influences physiological and biochemical processes such as growth, development, flowering, and senescence [141,201].

GB confers water stress tolerance to plants [207]. It is a quaternary ammonium compound with osmoprotective functions, and protects plants by stabilizing both the highly ordered quaternary structure of membranes and proteins [202]. The amount of GB increases under both saline and drought conditions. GB acts as a molecular chaperone and assists in enzyme folding and the recovery of protein functions. Inoculation of maize with *Klebsiella variicola* F2, *Raoultella planticola* YL2, and *P. fluorescens* YX2 increased choline and GB accumulation, which improved RWC and dry matter weight (DMW) under different stress regimes [208].

Inoculation of *Pseudomonas* SP2 increased the drought tolerance of *A. thaliana*; inoculated plants increased GB accumulation, proline, RWC, chlorophyll, and reduced MDA in shoots compared to the non-inoculated plant [202].

Although these studies showed that PGPR promote proline accumulation, other studies found that PGPR reduce proline accumulation [64,161,167,176,209]. These discrepancies can be attributed to the different mechanism of PGPR to combat drought, changes in bacterial species, bacterial communication mechanisms with plants, bacterial interaction, and severity of drought stress. Therefore, it can be concluded that the accumulation of osmoprotectants allows osmotic regulation of plants, thus mitigating drought stress.

In addition, trehalose is a non-reducing disaccharide. It plays an important role in maintaining the osmotic potential of plants by protecting biological structures from desiccation damage [29]. Studies have reported that high levels of trehalose have a positive impact on the survival and yield of bean (*Phaseolus vulgaris*) plants nodulated by rhizobial strain *Rhizobium etli* after severe and long periods of drought [210]. Vílchez et al. [211] also proved the protection of tomato and green bell pepper (*Capsicum annuum* L.) plants against drought by *P. putida* KT2440 that expresses otsAB genes for trehalose biosynthesis compared to untreated plants.

PGPR, genetically engineered to produce excess trehalose, can give plants drought endurance, as evidenced by modified *R. etli* and bean plant response showing more nodules, higher biomass production, and better recovery from drought stress in plants treated with modified *R. etli* than those treated with the wild type [212].

*4.4. Exopolysaccharide Production*

Exopolysaccharides (EPS) play an important role in mitigating water stress for microbial and plant populations. Bacteria form EPS as a survival strategy under adverse conditions. Bacterial EPS have been widely studied in the rhizosphere for their role in water retention. They have been shown to help hydrate plants and increase water and nutrient uptake from the soil and can enhance plant growth, development, and survival [135,163].

EPS form an attachment zone between soil particles, root systems, and bacteria that increases soil aggregation and colonization process, and maintains a long-term relationship with the host plant [155,213]. Plants treated with EPS-producing bacteria show increased resistance to water stress. It has been postulated that EPS provides a microenvironment that retains water and dries out more slowly than the surrounding environment, thus protecting bacteria and plant roots from desiccation [135].

EPS produced by PGPRs promote permeability by improving soil aggregation and maintaining a higher water potential around the roots. This increases nutrient uptake and stimulates plant development and water stress tolerance [69,135].

Another interesting feature of microbial water stress alleviation is the improvement in the adherent soil to root tissue ratio (ASR/RT) following inoculation with EPS-producing bacteria. It has been postulated that higher EPS content and improved RAS aggregation may help plants absorb a greater volume of water and nutrients from the rhizospheric soil, resulting in improved plant growth; in addition, this phenomenon is also known to counteract the negative effects of water stress [199]. Exopolysaccharides form hydrophilic biofilms that provide protection against aridity during osmotic stress by increasing the water retention potential of the soil and regulating the distribution of biological carbon sources [213].The inoculation of sunflower with *Pseudomonas* GAP-P45 producing a large amount of EPS improved germination, plant growth by increasing plant biomass, and RAS/RT ratio under water stress compared to non-inoculated seedlings. EPS production allowed better colonization by the stump, increased the volume of soil attached to the root, and improved the stability of soil aggregates [199]. Another study by Ilyas et al. [69] showed the fundamental role of EPS production in drought tolerance and osmotic stress. Co-inoculation with *B. subtilis* and *A. brasilense* produced a large amount of EPS and, with the production of phytohormone (IAA, ABA, GA), increased the germination rate of wheat seeds' seedling vigor index. In addition, it improves osmotic potential, water potential, and chlorophyll content (Chl a, b, and carotenoid), osmolyte content (proline, sugars, and proteins), and causes a significant increase in antioxidant enzymes (SOD, CAT, and POD), conferring better drought resistance [69].

Inoculation of *P. azotoformans* FAP5 strain generating EPS increased the germination rate of wheat seeds under PEG concentrations, generating osmotic stress [64]. Recently, a study by [163] showed improved germination rate, colonization, and growth parameters (root diameter, root area, root and shoot length, and seedling dry biomass) in wheat plants inoculated with *Klebsiella* sp. LEW16 under water-limited conditions due to different polyethyleneglycol (PEG) concentrations.

### 4.5. Modification of Phytohormone Activity by PGPR under Drought

The production of phytohormones by PGPR is considered one of the most important mechanisms by which many rhizobacteria promote plant growth. Auxins, GAs, ABA, and CKs are among the major groups of phytohormones that help plants tolerate abiotic and biotic stresses. Under drought stress conditions, PGPR promote plant growth through the production and modification of plant growth regulators and phytohormones [88,146].

IAA is the most abundant phytohormone and is produced generally by 80% of PGPR species [10,57]. IAA produced by bacteria can loosen plant cell walls, increasing the amount of root exudates, which further supports the growth of rhizosphere microorganisms and the colonization process [110,135]. Plants inoculated with IAA-producing bacteria showed greater tolerance to drought stress [46]. They showed an increase in shoot biomass and changes in the root architecture, increasing mineral nutrition and water interactions [15,110].

Pereira et al. [214] showed that co-inoculation of maize plants under drought conditions with two osmotolerant and IAA-producing PGPR, *Cupriavidus necator* 1C2 and *P. fluorescens* S3X, increased plant biomass and also increased nitrogen (N) and phosphorus (P) use efficiency. Under low moisture content (55% field capacity), the bacterial consortium (*Mesorhizobium cicero*, *B. subtilis*, and *B. mojavensis*) presenting IAA increased shoot and root biomass and improved water stress tolerance in chickpea compared to non-inoculated plants [168].

In addition, PGPR can regulate the expression of auxin signaling or transport synthesis genes in plants. Inoculation of *B. altitudinis* FD48 altered the architecture of the rice root system, which was established by examining the expression pattern of genes encoding defense responding to primary root and lateral root formation. Inoculation of rice with *B. altitudinis* FD48 altered the endogenous level of IAA by influencing the genes encoding

auxin expression, changing the architecture of the root system, and ensuring the survival of the plant under water stress [186].

Similarly, the study by Barnawal et al. [13] suggests that PGPR confer water stress tolerance to wheat plants by increasing IAA content through an indirect mechanism: by reducing ABA/ACC (the ethylene precursor) content and modulating the expression of a regulatory component (CTR1) of the (CTR1) ET signaling pathway.

ABA is a hormone commonly referred to as stress hormone that further induces physiological changes in the plant and modulates its growth [57,110]. Plants inoculated with PGPR showing high concentration of ABA have an improved ability to tolerate drought. ABA is involved in stomatal regulation by regulating leaf transpiration rate and root hydraulic conductivity or upregulating AQP [1,69,110]. Asghari et al. [201] reported that inoculation of *A. brasilense* increased ABA accumulation in pennyroyal mint (*Mentha pulegium* L.) plants under drought and watered conditions compared to non-inoculated plants. Thus, ABA accumulation may increase root hair elongation and regulate auxin transport in plant root tips [215]. Soybean plants inoculated with *P. simiae* Au under water-limited conditions produced more ABA and SA and less ET, resulting in improved drought tolerance compared to non-inoculated plants [216].

However, the regulation of hormones under stress conditions is a complex phenomenon. Other studies have reported a decrease in ABA content in plants treated with PGPR; Curá et al. [217] showed that maize plants inoculated with *Herbaspirillum seropedicae* Z-152 and *A. brasilense* SP-7 presented an improvement in plant biomass and drought tolerance with lower levels of ABA and ethylene in comparison with non-inoculated plants. Another study by Zhang et al. [155] also showed a reduction in ABA accumulation, an increase in growth parameters, and IAA accumulation in Jujube (*Ziziphus jujuba* L.) plants inoculated with *P. lini* DT6 and *Serratia plymuthica* DT8 compared to non-inoculated plants.

Drought stress causes a reduction in CK concentration [95]. The use of PGPR can increase CK concentrations under water-limited conditions. Selvakumar et al. [179] showed that inoculation of tomato plants with CK-producing bacteria *Citricoccus zhacaiensis* and *B. amyloliquefaciens* improved photosynthesis, transpiration, relative water content, and yield of plants during water stress.

GAs are plant hormones that regulate a number of physiological functions in plants at different stages of development [57,151]. GA-producing bacteria have been shown to increase the drought resistance of various plants. GAs produced by *P. putida* H-2-3 increased root system growth and soybean yield, and induced regulation of stress hormones and antioxidants under drought conditions [218]. A study by Cohen et al. [219] showed that *A. lipoferum*-producing GA and ABA improved the drought tolerance of maize plants.

Other plant hormones such as JA and SA can protect plants from oxidative stress damage. Several bacteria alter the amount of these hormones in the plant under stress [220]. Similarly, JA are involved in increasing drought tolerance in plants, and it has been reported that exogenous application of JA increases the production of various antioxidants that ultimately contribute to drought reduction [1,220].

### 4.6. 1-aminocyclopropane-1-carboxylic Acid (ACC) Deaminase

One of the main strategies used by PGPR to mitigate ethylene stress is the hydrolysis of 1-aminocyclopropane-1-carboxylic acid (ACC) by the enzyme ACC deaminase. Decreased levels of ACC lead to decreased endogenous ethylene levels and their detrimental effects on plants [26,221]. Research has shown that PGPR possessing ACC deaminase can cleave the precursor ACC and hydrolyze it into ammonia and $\alpha$-ketobutyrate that are used as carbon and nitrogen sources by bacteria [222,223]. ACC deaminase (EC 3.5.99.7) is a pyridoxal $5'$-phosphate (PLP)-dependent multimeric enzyme found in the cytoplasm of bacteria [224,225]. Plant drought tolerance has been linked to bacterial upregulation of the ACC deaminase (acdS) gene, which cleaves the ethylene precursor (ACC). In addition, putative ACC deaminase (acdS) genes have been widely found and described in various bacterial genomes. Some bacteria, such as *A. fragmenthaudii, B. licheniformis, Variovorax*

*paradoxus*, *Pseudomonas* spp., *B. phytofirmans*, and *Enterobacter* spp. have been reported to increase plant drought resistance through the production of this enzyme [225,226].

*Variovorax paradoxus* RAA3, with its high capacity for ACC deaminase production as well as $N_2$ fixation and siderophore production capabilities, significantly improved wheat plant growth and leaf nutrient concentrations, and activated antioxidant systems and osmolyte accumulation [226]. Inoculation of wheat with *Agrobacterium fabrum* and *B. amyloliquefaciens* with ACC deaminase activity under water stress resulted in improved morphological parameters (lateral root area and length), increased nutrient uptake (N, P, K), photosynthetic rate, and photosynthetic pigments compared to the non-inoculated plant [226]. A study by Dubey et al. [153] on inoculation of soybean with three endophytic strains *B. cereus* AKAD A1-1, *P. otitidis* AKAD A1-2, and *Pseudomonas* sp. AKAD A1-3 proved that these ACC deaminase-producing strains effectively alleviated drought stress through various mechanisms: by improving plant growth, membrane integrity, relative water content, and accumulation of compatible solutes and osmolytes. Treatment of wheat seeds with ACC- and EPS-producing strains *Pseudomonas* sp. and *S. marcescens* improved plant water status, chlorophyll, carotenoid content, and proline accumulation, confirming the oxidative stress tolerance and drought resistance compared to untreated seeds [152].

### 4.7. Volatile Organic Compounds (VOCs)

Research has shown that PGPR, in addition to root colonization, can effectively improve tolerance to abiotic stress without physical contact with host plants but rather through the release of chemicals called volatile organic compounds (VOCs). VOCs are secondary metabolites produced by bacteria which act as signaling molecules facilitating short- and long-distance transmission, as well as inter- and intra-organism communication. They are also responsible for activating various signaling pathways in plants to trigger induced systemic resistance (ISR), which enhances plant defense mechanisms against abiotic and biotic stress [227]. Bhattacharyya et al. [228] proved a significant increase in fresh weight, root and shoot length, number of lateral leaves, and leaf area of *A. thaliana* plants exposed to VOCs produced by *Proteus vulgaris* JBLS2020. Exposure to the microbial volatile organic molecule 2R,3R-butanediol produced by *P. chlororaphis* O6 promoted drought resistance in *A. thaliana* seedlings through the induction of stomatal closure by reducing water loss compared to untreated plants [229]. Similarly, a study by Yasmin et al. [202] showed that VOCs (dimethyl disulfide, 2,3-butanediol and 2-pentylfuran) from *P. pseudoalcaligenes* stimulate plant growth and induce drought tolerance in maize (*Zea mays* L.). Plants treated with VOCs showed better plant growth and reduced MDA levels and electrolyte loss compared to untreated plants.

### 4.8. Modification of the Plant Antioxidant Defense System by PGPR

ROS cause oxidative damage, thus affecting the metabolic processes of plant cells and can cause cell death [230]. Several studies have proved that inoculation with PGPR help plants overcome oxidative stress of ROS by increasing the activities of enzymatic and non-enzymatic antioxidant systems, which results in a decrease in MDA accumulation, electrolyte leakage, and $H_2O_2$ content, and increases the stability of membranes and physiological processes of plants [66,151].

According to Rachid et al. [174], inoculation of wheat with *B. megaterium* under water-limited conditions increased the activities of antioxidant enzymes SOD, APX, POD, CAT, and GR by 159, 87, 164, 166 and 60%, respectively, in comparison with non-inoculated plants. Inoculation reduced oxidative stresses by decreasing MDA content by 57% and electrolytes by 33%, and increasing the process of photosynthesis compared to non-inoculated plants. This confirms that PGPR interfere with physiological processes and antioxidant defense systems to improve drought tolerance.

A study by He et al. [66] reported an increase in the activities of ROS scavenging enzymes including CAT, POD, and SOD in ryegrass (*Lolium perenne* L.) plants inoculated with *Bacillus* sp. WM13-24 and *Pseudomonas* sp. M30- 35 under water shortage conditions

compared to non-inoculated plants. In addition, the decrease in MDA content, relative membrane permeability (RMP), and $H_2O_2$ accumulation in inoculated plants allows us to conclude that there is a link between RMP inoculation, increased antioxidant enzyme activities, and drought tolerance.

Similarly, inoculation of common myrtle (*Myrtus communis* L.) with *P. fluorescens* and *P. putida* improved plant growth and biomass, chlorophyll and carotenoid content, and stimulation of enzymatic and non-enzymatic defenses (Phenol and flavonoids) [137].

### 4.9. Molecular Studies in Drought Stress Mitigation by PGPR/Alteration of Stress Responsive Gene Expression

Many plants exhibit PGPR-mediated stress resistance via the activation of numerous genes in response to abiotic stressors. Several genomic expression studies using molecular biology techniques have shown that stressed plants inoculated with PGPR exhibit different gene expression profiles than those not inoculated [231,232].

Li et al. [141] proved that treating drought-stressed wheat with a cell filtrate of *Streptomyces* pactum Act12 improved plant growth and development. In addition, *S. pactum* Act 12 strain induced molecular responses by upregulating the expression levels of genes related to water deficit resistance. Therefore, Act12 treatment allowed cell wall expansion by inducing EXPA2 and EXPA6 expression. Overexpression of EXPA2 was favorable for proline accumulation and ROS removal in wheat and reduced MDA content in plants. Act12 treatment significantly increased the expression of the P5CS gene that regulates proline content in plants during early water stress. Drought also induced increased expression of the SnRK gene that encodes a protein kinase promoting leaf stomata closure, but also promotes root formation and enhances both osmotic adjustment capacity and photosynthesis in plants [90].

Bacterial colonization by PGPR strains can significantly alter gene expression in stressed plants. During drought stress, rice plants inoculated with the *P. fluorescens* (Pf1) strain showed the activation of six genes: COX1 (regulates energy and carbohydrate metabolism), PKDP, AP2-EREBP, Hsp20, bZIP1, and COC1. Hsp20, bZIP1, and COC1 are chaperones involved in the ABA signaling pathway; PKDP is a protein kinase; and AP2-EREBP is involved in developmental and stress defense pathways [231].

Inoculation with *Arthrobacter protophormiae* SA3, *Dietzia natronolimnaea* STR1, and *B. subtilis* LDR2 confers drought and salinity tolerance in wheat plants by altering the ethylene signaling pathway regulatory gene and transcription factor expression [13].

Using sequencing Illumina, researchers found that inoculation of sugarcane (*Saccharum* spp.) with *Gluconacetobacter diazotrophicus* PAL5 under water stress circumstances activated ABA-dependent signaling genes [233]. Moreover, the use of Illumina HiSeq 2500 revealed that the expression of genes of maize inoculated with *P. putida* FBKV2 under water-limited conditions were modified. On the one hand, there was a downregulation of ET biosynthesis, ABA and auxin signaling, SOD, CAT, and POD in seedlings inoculated with *P. putida* FBKV2. On the other hand, genes involved in alanine and choline biosynthesis, heat shock proteins, and LEA proteins were induced to play a role in drought tolerance [234]. *Arabidopsis thaliana* plants inoculated with GAP-P45 revealed a change in the expression patterns of genes associated with polyamine biosynthesis (ADC, AIH, CPA, SPDS, SPMS, and SAMDC), thereby increasing the levels of polyamines in plants, and mitigating osmotic stress during drought [235].

Woo et al. [190] also showed that inoculation of *A. thaliana* and *B. campestris* with *B. subtilis* GOT9 resulted in improved tolerance and survival under drought conditions via hyper-induction of various genes (RD29B, RAB18, RD20, and NCED3) in *A. thaliana* and (BrDREB1D (Bra028290), BrWRKY7 (Bra013732), BraCSD3 (Bra002100), and BraCSD3 (Bra002133)) in *B. campestris*, responsible for the GOT9-induced enhancement of drought tolerance phenotypes.

Quantitative real-time (qRT)-PCR analysis showed differential expression of genes involved in transcription activation (DREB1A and NAC1), stress response (LEA and DHN),

ROS scavenging (CAT, APX, GST), ethylene biosynthesis (ACO and ACS), salicylic acid (PR1), and jasmonate (MYC2) signaling in two chickpea plants exposed to drought stress. Inoculation with *P. putida* (RA) confers drought tolerance in chickpea by modulating the differential expression of stress-responsive genes [176].

Similarly, Vaishnav and Choudhary [216], by using qRT-PCR, showed that under drought, upregulation of genes DREB/EREB encoding transcription factors, P5CS; GOLS for osmoprotectants, and PIP; TIP for water transporters (Aquaporin) in soybean inoculated with *P. simiae* AU resulted in improved drought tolerance compared to non-inoculated plants.

In addition, inoculation of *M. oleifera* with *B. pumilus* increased leaf folate, tocopherol, and carotenoid levels by upregulating genes involved in γ-tocopherol methyltransferase (γ-TMT), lycopene cyclase (LBC), phytoene desaturase (PDS) and phytoene synthase (PSY), and dihydrofolate reductase thymidylate synthase (DHFR-TS) [236].

Recently, by using qRT- PCR, Omara et al. [237] proved that inoculation of rice plants with *B. megaterium*, *P. azotoformans*, and *Rhizobium* sp. resulted in improved plant growth and drought tolerance by altering the expression of growth and stress response-related genes (COX1, AP2-EREBP, GRAM, NRAMP6, NAM, GST, and DHN) and expansion genes (EXP1, EXP2, and EXP3). The expression of these genes was strongly induced in plants inoculated with *B. megaterium* and was correlated with an improved growth status and tolerance under drought stress.

## 5. Conclusions and Perspectives

Drought has been identified as the main constraint causing the decline in agricultural production. Agriculture and water resources are currently going through an unprecedented critical period on an international scale. Drought is directly proportional to climate change. Drought stress is responsible for the alteration of plant growth but also of soil microbial interactions. Thus, the reduction in agricultural production in the world is a serious threat to sustainable agriculture. Recent studies have proven the importance of using drought tolerant PGPRs to improve plant growth and drought tolerance. This review has attempted to synthesize the variety of drought-adaptable microorganisms and the understanding of the mechanisms used by these bacteria to promote plant tolerance down to the molecular level. Future research should focus on identifying more advantageous bacterial strains that could be exploited to develop bio-inoculants. In addition, the molecular processes underlying microorganism-induced abiotic stress tolerance in plants need to be investigated. Therefore, new molecular research on plant–microbe interactions is needed in the future to better understand the pathways used by rhizospheric microorganisms in induced systemic tolerance and rhizospheric engineering under drought stress. Thus, new research should focus on bio formulations, such as the encapsulation of strains to ensure the efficacy of bio-inoculants under field conditions. Improving drought tolerance, colonization, and dissemination of beneficial bacteria is a strategy with the ultimate goal of food security. Understanding the interactions between plants, associated microorganisms, and the environment is a possible method to mitigate the negative consequences of climate change on crop productivity and sustainability.

**Author Contributions:** Conceptualization, L.B., H.C.-S. and A.S.; methodology, N.B., A.C.B., L.B., H.C.-S., A.S. and O.B.; software, A.C.B., L.B.; H.C.-S., A.S. and O.B.; validation, L.B., H.C.-S. and A.S.; formal analysis, N.B., A.C.B., L.B., H.C.-S. and O.B.; investigation, A.C.B., L.B., H.C.-S., A.S. and O.B.; resources, A.C.B., L.B., H.C.-S. and A.S.; data curation, N.B., H.C.-S., A.S., A.C.B., L.L., F.N.A., O.B. and L.B.; writing—original draft preparation, N.B., H.C.-S., A.S. and L.B.; writing—review and editing, N.B., H.C.-S., A.S., A.C.B., L.L., F.N.A., O.B. and L.B.; visualization, A.C.B. and L.B.; supervision, L.B., H.C.-S. and A.S.; project administration, L.B., H.C.-S. and A.S.; funding acquisition, L.B., H.C.-S. and A.S. All authors have read and agreed to the published version of the manuscript.

**Funding:** This research received no external funding.

**Institutional Review Board Statement:** Not applicable.

**Informed Consent Statement:** Not applicable.

**Data Availability Statement:** Not applicable.

**Conflicts of Interest:** The authors declare no conflict of interest.

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
