# Peer review of "Plant Growth-Promoting Rhizobacteria (PGPR): A Rampart against the Adverse Effects of Drought Stress"

_water, doi:10.3390/w15030418_

Round 1

Reviewer 1 Report

The review attempts to provide a comprehensive information about the roles of PGPRs on plant tolerance to drought.  However, the authors do not seem to have a clear understanding about the concept of plant water relationship and photosynthesis.  Several mistakes were found in the manuscript regarding the concept of water potential dan photosynthesis. A comprehensive discussion regarding the physiological and molecular mechanisms of the PGPR roles to drought tolerance of plants needs to be added in the review. Comments and suggestions are provided in the file.

Author Response

Thank you very much for making a critical assessment of our manuscript. Based on the comments, we submit a revised version, which takes into account all of the points raised by the reviewer(s). Our response to reviewers’ comments has been listed below point-by-point.

Reviewer 1

1 62-65 Please delete the sentence

Correction has been done.

2 66 Word "Therefore" should not be put in the begining of a senctence or a paragraph

Correction has been done.

3 81-83 This sentence has a negative tone to the strategies of increasing plant tolerance to drought which have been mentioned by the authors in lines 79-81. Because the strategy offered by the authors is a complement to other strategies, this sentence should be rewritten so that it has a more positive tone

Correction has been done.

4 84 Please remove word: "more powerful and"

Correction has been done.

5 93 Please replace word: "plant resistance" with "plant tolerance" 6 Tabel 1 Please replace word: "Increasement" with "Increase of" Please adjust the font size to the same font size thorough the table.

Correction has been done.

7 118-144 In this sub-title, the authors fail to correctly describe the effect of drought stress to nutrient uptake by plant roots. It seems that the authors do not well understand on the concept of plant water potential, plant mineral nutrition, and nutrient uptake. The sub-themes of this section also do not focus on discussing the sub-topic of the effect of drought stress on nutrition uptake. Please rewrite the section.

Correction has been done.

8 145-159 This section was not well written. Some sentences are incomplete. Several words are inappropriately chosen. The content of the section does not support the sub-title idea. Please rewrite the section.

Correction has been done.

9 160-188 The authors fail to correctly describe the concept of photosynthesis in relation to the role of water, stomata, and gas exchange, especially the direct and indirect effects of water deficit to photosynthesis transport electron, carbon fixation, and yield. An important molecule in Calvin cycle, Ribulosa-1,5-bisphosphate, was incorrectly written (lines 178-179). The authors are also inaccurately correlate between chlorophyll, leaf area, photosynthesis, drought tolerance, and yield. This section was not systematically written. Please rewrite the section.

Correction has been done.

10 189-212 In this section, the author tries to relate drought stress to the balance of phytohormones in plants. However, the narrative in this section is written incoherently, making it difficult to follow the storyline of this section. The author seems to only combine sentences without a clear storyline. For example, the sentence in lines 206-208, it suddently appears in that place without clear meaning and connection of the sentence with the paragraph idea.

Correction has been done.

11 229-231 The sentence in lines 229-231: "Thus, Membrane stability affected by water stress was detected by measuring electrolyte leakage (EL) from the cell" is not necessary to be in this paragraph.

Correction has been done.

12 231-232 The sentence needs to be corrected in term of drought disrupts lipid binding to membrane protein. The main cause of membrane instabiltiy is due to lipid peroxidation caused by reactive oxygen species produced by drought stress.

Correction has been done.

13 Figure 1 Figure 1 needs to be reconstructed to make it clearer the connectivity among the components in drought stress and plant responses

Correction has been done.

14 286-289 The sentence in these lines indicate that the authors has misconception on osmotic potential as the component of cell water potential. Osmotic adjustment that occurs during drought stress is to decrease osmotic potential (to be more negative), not to improve osmotic pressure. By decreasing osmotic potential, the cell water potential will be maintained below the soil water potential allowing water to enter the cell, so the cells are able to maintain cell water potential in certain level.

Correction has been done.

15 296-297 The sentence is not clear

Correction has been done.

16 303-304 The statement in this sentence is questionable. The cited literature does not state as the authors stated in this tentence.

Correction has been done.

17 Figure 2 Figure 2 need to be corrected to provide better and correct illustration, such as word "apoplast" seems to be misplaced.

Correction has been done.

18 577-581 The statements in these lines need to rewrite to provide correct information and clearer idea.

Correction has been done.

19 598-603; 660-664 When PGPRs secret osmolytes or EPS in the rhizosphere, it will increase solute in the soil water producing lower osmotic potential, which is consequently it will decrease soil water potential that can be lower than cell water potential. In other word, it can generate more severe drought stress. Please clarify.

Correction has been done.

20 857-859 Please rewrite

Correction has been done.

21 863 Where does the word "PRMPs" come from?

Correction has been done.

22 all documents Scientific names of the species must be written in italics

Correction has been done.

23 Discussion Instead of revieweing the phenomena of plant responses to PGPR in relation to drought stress, please provide a comprehensive discussion of the physiological and molecular mechanisms regarding the role of PGPR in increasing plant tolerance to drought

Correction has been done.

Reviewer 2 Report

Dear Authors,

                  Although the manuscript titled "Plant Growth Promoting Rhizobacteria (PGPR) a rampart 2 against the adverse effects of drought stress" is written well with significant information collected, however adding more updated information will enhance further the value of review paper. I would suggest to add more recent citation by replacing the old ones.  

Author Response

Reviewer 2

Although the manuscript titled "Plant Growth Promoting Rhizobacteria (PGPR) a rampart 2 against the adverse effects of drought stress" is written well with significant information collected, however adding more updated information will enhance further the value of review paper. I would suggest to add more recent citation by replacing the old ones.  

Correction has been done.

Reviewer 3 Report

In this review, the authors systematacially reviewed the effects of Plant Growth Promoting Rhizobacteria (PGPR) on different species from morphological, physiological, biochemical and molecular levels under drought stress condition, which provides a theoretical basis for the effective use of PGPR in future. However, some defects in this MS still need to be modified.

As following:

1.       Both the strain’s and specie’s Latin names should be italic, please check and revise throughout the MS.

2.       Line 616: the citation is wrong, and this sentence contains a tense error.

3.       Each quotation in MS should be in the same tense.

4.       In this review, the authors paid more attention to summarize the morphological, physiological, biochemical changes of PGPR inoculated species, while the molecular regulation mechanisms of PGPR involved in drought tolerance of different species are rarely referred. Therefore, if possible, I suggest that the authors supplement the exited researches on the molecular regulatory mechanisms of PGPR involved in drought tolerance of different species.

5.       The citations and references format need meet to the journal’s request.

Author Response

Reviewer 3

  1. Both the strain’s and specie’s Latin names should be italic, please check and revise throughout the MS.

Correction has been done.

  1. Line 616: the citation is wrong, and this sentence contains a tense error.

Correction has been done.

  1. Each quotation in MS should be in the same tense.

Correction has been done.

  1. In this review, the authors paid more attention to summarize the morphological, physiological, biochemical changes of PGPR inoculated species, while the molecular regulation mechanisms of PGPR involved in drought tolerance of different species are rarely referred. Therefore, if possible, I suggest that the authors supplement the exited researches on the molecular regulatory mechanisms of PGPR involved in drought tolerance of different species.

Correction has been done.

  1. The citations and references format need meet to the journal’s request.

Correction has been done.